# ObjectNet: A large-scale bias-controlled dataset for pushing the limits of object recognition models

**Andrei Barbu**[*]
MIT, CSAIL & CBMM

**David Mayo**[*]
MIT, CSAIL & CBMM

**Julian Alverio**
MIT, CSAIL

**William Luo**
MIT, CSAIL

**Christopher Wang**
MIT, CSAIL

**Dan Gutfreund**
MIT-IBM Watson AI

**Joshua Tenenbaum**
MIT, BCS & CBMM

**Boris Katz**
MIT, CSAIL & CBMM

## Abstract

We collect a large real-world test set, ObjectNet, for object recognition with controls where object backgrounds, rotations, and imaging viewpoints are random. Most scientific experiments have controls, confounds which are removed from the data, to ensure that subjects cannot perform a task by exploiting trivial correlations in the data. Historically, large machine learning and computer vision datasets have lacked such controls. This has resulted in models that must be fine-tuned for new datasets and perform better on datasets than in real-world applications. When tested on ObjectNet, object detectors show a 40-45% drop in performance, with respect to their performance on other benchmarks, due to the controls for biases. Controls make ObjectNet robust to fine-tuning showing only small performance increases. We develop a highly automated platform that enables gathering datasets with controls by crowdsourcing image capturing and annotation. ObjectNet is the same size as the ImageNet test set (50,000 images), and by design does not come paired with a training set in order to encourage generalization. The dataset is both easier than ImageNet – objects are largely centered and unoccluded – and harder, due to the controls. Although we focus on object recognition here, data with controls can be gathered at scale using automated tools throughout machine learning to generate datasets that exercise models in new ways thus providing valuable feedback to researchers. This work opens up new avenues for research in generalizable, robust, and more human-like computer vision and in creating datasets where results are predictive of real-world performance.

## 1 Introduction

Datasets are of central importance to computer vision and more broadly machine learning. Particularly with the advent of techniques that are less well understood from a theoretical point of view, raw performance on datasets is now the major driver of new developments and the major feedback about the state of the field. Yet, as a community, we collect datasets in a way that is unusual compared to other scientific fields. We rely almost exclusively on dataset size to minimize confounds (artificial correlations between the correct labels and features in the input), to attest unusual phenomena, and encourage generalization. Unfortunately, scale is not enough because of rare events and biases – Sun et al. [1] provide evidence that we should expect to see logarithmic performance increases as a function of dataset size alone. The sources of data that datasets draw on today are highly biased, e.g., object class is correlated with backgrounds [2], and omit many phenomena, e.g., objects appear in stereotypical rotations with little occlusion. The resulting datasets themselves are similarly biased [3].

---

[*]Equal contribution. Website https://objectnet.dev. Corresponding author abarbu@csail.mit.edu

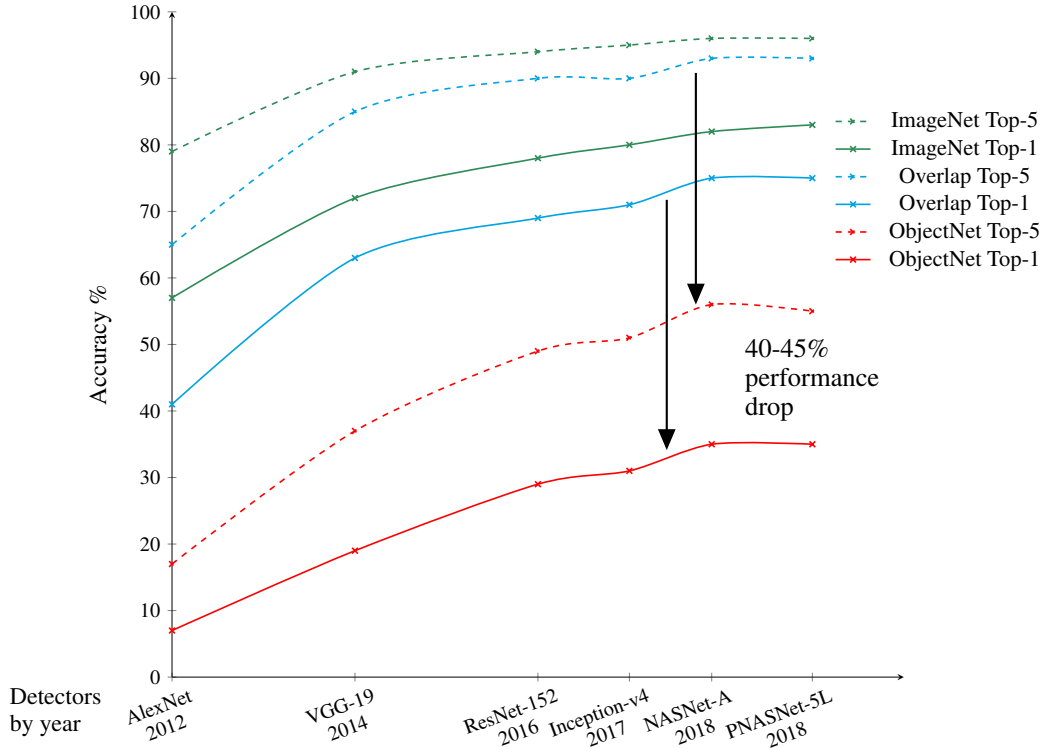

Figure 1: Performance on ObjectNet for high-performing detectors trained on ImageNet in recent years: AlexNet [4], VGG-19 [5], ResNet-152 [6], Inception-v4 [7], NASNET-A [8], and PNASNet-5 Large [9]. Solid lines show top-1 performance, dashed lines show top-5 performance. ImageNet performance on all 1000 classes is shown in green. ImageNet performance on classes that overlap with ObjectNet is shown in blue; the two overlap in 113 classes out of 313 ObjectNet classes, which are only slightly more difficult than the average ImageNet class. Performance on ObjectNet for those overlapping classes. We see a 40-45% drop in performance. Object detectors have improved substantially. Performance on ObjectNet tracks performance on ImageNet but the gap between the two remains large.

In other areas of science, such issues are controlled for with careful data creation and curation that intentionally covers phenomena and controls for biases – important ideas that do not easily scale to large datasets. For example, models for natural language inference, NLI, that perform well on large datasets fail when systematically varying aspects of the input [10], but these are not collected at scale. In computer vision, datasets like CLEVR [11] do the same through simulation, but simulated data is much easier for modern detectors than real-world data. We show that with significant automation and crowdsourcing, you can have scale and controls in real-world data and that this provides feedback about the phenomena that must be understood to achieve human-level accuracy.

ObjectNet is a new large crowdsourced test set for object recognition that includes controls for object rotations, viewpoints, and backgrounds. Objects are posed by workers in their own homes in natural settings according to specific instructions detailing what object class they should use, how and where they should pose the object, and where to image the scene from. Every image is annotated with these properties, allowing us to test how well object detectors work across these conditions. Each of these properties is randomly sampled leading to a much more varied dataset.

In effect, we are removing some of the brittle priors that object detectors can exploit to perform well on existing datasets. Overall, current object detectors experience a large performance loss, 40-45%, when such priors are removed; see fig. 1 for performance comparisons. Each of the controls removes a prior and degrades the performance of detectors; see fig. 2 for sample images from the dataset. Practically, this means that an important feedback for the community about the limitations of models is missing, and that performance on datasets is limited as a predictor of the performance users can expect on their own unrelated tasks.

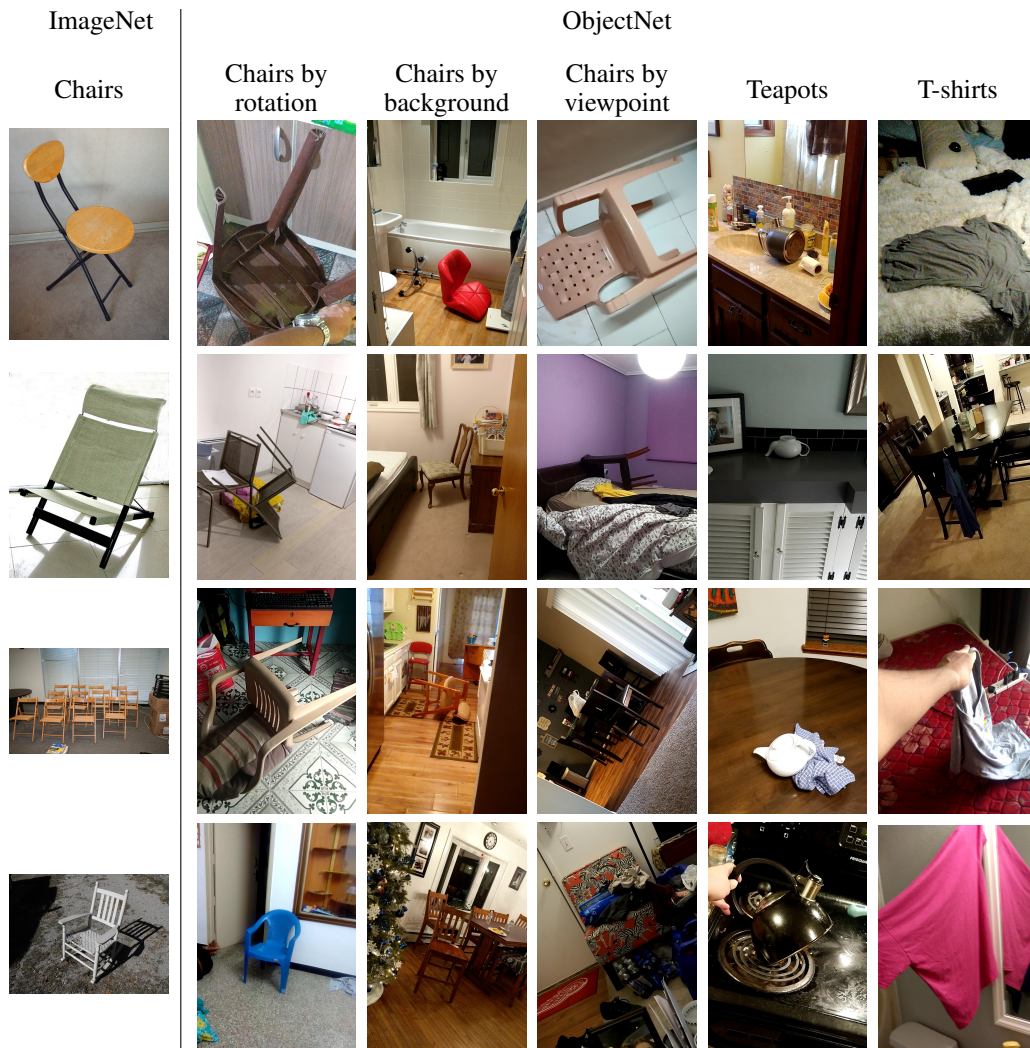

Figure 2: ImageNet (left column) often shows objects on typical backgrounds, with few rotations, and few viewpoints. Typical ObjectNet objects are imaged in many rotations, on different backgrounds, from multiple viewpoints. The first three columns show chairs varying by the three properties that are being controlled for: rotation, background, and viewpoint. One can see the large variety introduced to the dataset because of these manipulations. ObjectNet images are lightly cropped for this figure due to inconsistent aspect ratios. Most detectors fail on most of the images included in ObjectNet.

To encourage generalization, we make three other unusual choices when constructing ObjectNet. First, ObjectNet is only a test set, and does not come paired with a training set. Separating training and test set collection may be an important tool to avoid correlations between the two which are easily accessible to large models but not detectable by humans. Since humans easily generalize to new datasets, adopting this separation can encourage new machine learning techniques that do the same. Second, while ObjectNet will be freely available, it comes with an important stipulation: one cannot update the parameters of any model for any reason on the images present in ObjectNet. While fine-tuning for transfer learning is common, it encourages overfitting to particular datasets – we disallow fine-tuning but report such experiments in section 4.3 to demonstrate the robustness of the dataset. Third, we mark every image by a one pixel red border that must be removed on the fly before testing. As large-scale web datasets are gathered, there is a danger that data will leak between the training and test sets of different datasets. This has already happened, as Caltech-UCSD Birds-200-2011, a popular dataset, and ImageNet were discovered to have overlap putting into question some results [12]. With test set images marked by a red border and available online, one can perform reverse image search and determine if an image is included in any training set anywhere. We encourage all computer vision datasets – not just ones for object detection – to adopt this standard.

While it includes controls, ObjectNet is not hard in arbitrary ways. It is in many ways intentionally easy compared to ImageNet or other datasets. Objects are highly centralized in the image, they are rarely occluded and even then lightly so, and many backgrounds are not particularly cluttered. In other senses, ObjectNet is harder, a small percentage of viewpoints, rotations, and even object instances, are also difficult for humans. This demonstrates a much wider range of difficulty and provides an opportunity to also test the limits of human object recognition – if object detectors are to augment or replace humans, such knowledge is critical. Our overall goal is to test the bias of detectors and their ability to generalize to specific manipulations, not to just create images that are difficult for arbitrary reasons. Future versions of the dataset will ratchet up this difficulty in terms of clutter, occlusion, lighting, etc. with additional controls for these properties.

Our contributions are:

1. a new methodology to evaluate computer vision approaches on datasets that have controls,
2. an automated platform to gather data at scale for computer vision,
3. a new object recognition test set, ObjectNet, consisting of 50,000 images (the same size as the ImageNet test set) and 313 object classes, and
4. an analysis of biases at scale and the role of fine-tuning.

## 2 Related work

Many large datasets for object recognition exist such as ImageNet [13], MS COCO [14], and OpenImages [15]. While the training sets for these datasets are huge, the test sets are comparable to the size of the dataset presented here, with ImageNet having 50,000 test images, MS COCO having 81,434, and OpenImages having 125,436, compared to ObjectNet's 50,000 test images. Such datasets are collected from repositories of existing images, particularly Flickr, which consist of photographs – images that users want to share online. This intent biases against many object instances, backgrounds, rotations, occlusion, lighting conditions, etc. Biases lead simultaneously to models that do not transfer well between datasets [3] – detectors pick up on biases inside a dataset and fail when those biases change – and that achieve good performance with little fine-tuning on new datasets [16] – detectors can quickly acquire the new biases even with only a few training images per class. In computer vision applications, biases may not match those of any existing dataset, they may change over time, adversaries may exploit the biases of a system, etc.

The dataset-dependent nature of existing object detectors is well-understood with several other approaches – aside from scale – having been attempted to alleviate this problem. Some focus on the datasets themselves, e.g., Khosla et al. [17] subdivide datasets into partitions that are sufficiently different, something possible only if datasets have enough variety in them. Others focus on the models, e.g., Zhu et al. [2] train models that separate foregrounds and backgrounds explicitly to become more resilient to biases. Demonstrating the value of models that have robustness built into them by design requires datasets that control for biases – controls are not just a sanity check, they encourage better research.

Some datasets, such as MPII cooking [18], KITTI [19], TACoS [20], CHARADES [21], Something-Something [22], AVA [23], and Partially Occluded Hands [24] collect novel data. Explicitly collecting data is difficult, as evidenced by the large gap in scale between these datasets and those collected from existing online sources. At the same time, explicit instructions and controls can lead to more varied and interesting datasets. These datasets on the whole do not attempt to impose controls by systematically varying some aspect of the data – users are prompted to perform actions or hold objects but are not told how to do this or what properties those actions should have. Workers choose convenient settings and manners in which to perform actions leading to biases in datasets.

## 3 Dataset construction

ObjectNet is collected by workers on Mechanical Turk who image objects in their homes; see fig. 3. This gives us control over the properties of those objects while also ensuring that the images are natural. We asked workers to image objects in 4 backgrounds (kitchens, living rooms, bedrooms, washrooms), from 3 viewpoints (top, angled at 45 degrees, and side), and in 50 object rotations. Rotations were uniformly distributed on a sphere, after which nearby points were snapped to the equator and the poles. We found that workers are able to pose objects to within around 20 degrees of

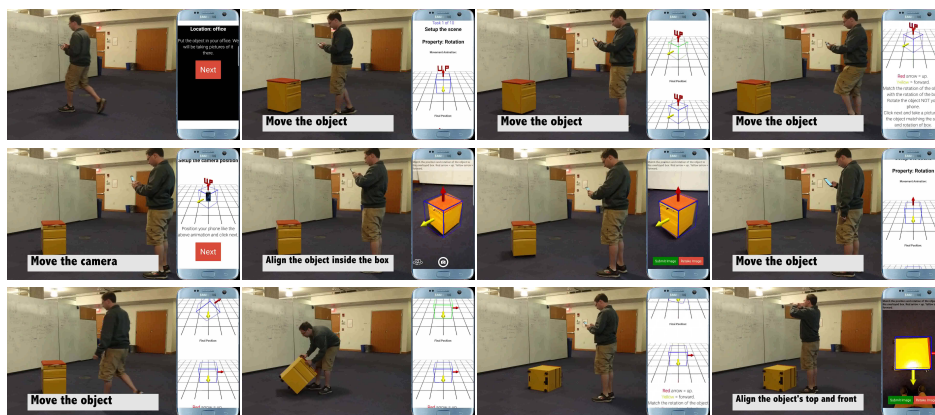

Figure 3: Workers select one object that they have available from a small number of choices. They are shown a rectangular prism, in blue, with two labeled orthogonal axes in red and yellow. These labels are object-class specific, so that workers can register the object correctly against the rectangular prism. We do not show workers images of desired objects to not bias them toward certain instances. Workers see an animation of how the object should be manipulated, perform this manipulation, and then align the object against the final rectangular prism rendered on their camera. Not shown above is the post-capture review UI to ensure that images contain the right objects and are not blurry.

rotation depending on the axis, although the uniformity of the resulting rotations varies by class. This could be more accurate, but we intentionally did not show instances of object classes to workers in order to avoid biasing them toward particular instances. In roughly one third of the trials we showed a rotated 3D car (cars do not appear in our dataset) as an additional cue for the desired rotation.

Workers are transitioned to their phone using a QR code, an object is described to them (but no example is shown), and they verify if an object that matches the description is available. A rectangular prism is then presented with labeled faces that are semantically relevant to that object, e.g., the front and top of a chair. Each object class was annotated with two semantically meaningful orthogonal axes, a single axis if the object class was rotationally symmetric, or no axis if it was spherical. We found that describing such parts in a manner that leads to little disagreement is difficult and requires careful validation. While this provides a weak bias toward particular object instances – one might imagine a chair with no distinctive front – it is necessary for explaining the desired object pose.

The rectangular prism is also animated to show the desired object pose. The animation starts with the rectangular prism representing the object in a default and common pose, e.g., the front of a chair facing a user and the top pointed upward, and then transitions it into the desired pose. Another animation shows the viewpoint from which the object should be imaged. We found that animating such instructions was critical in allowing workers to determine the desired object poses.

Workers are asked to move the object into a specific room, pose it, and image it from a certain angle. The rectangular prism was overlayed on their phone camera in the final desired position with the arrows marking the class-specific semantically-relevant faces. This also proved critical as remembering the desired rotation for an object is too unreliable.

This process annotates every image with three properties (rotation, viewpoint, and background); it controls for biases by sampling these properties randomly, thus allowing us to include objects in rotations and scenes that are unusual. Each image is validated to ensure that it contains the correct objects and that any identifying information is removed.

To select object classes for the dataset, we listed 420 common household objects. Of these, 55 classes were eliminated because they are not easily movable, e.g., beds (16 classes), pose a safety concern, e.g., fire alarms (8), were too confusing to subjects, e.g., we found little agreement on what armbands are (10), posed privacy concerns, e.g., people (5), or were alive and cannot be manipulated safely, e.g., plants (2); numbers do not add because classes were excluded for multiple reasons. In addition, 52 object classes were too rare, e.g., golf clubs. Data was collected for 313 object classes, with ≈160 images per class on average with a standard deviation of 44.

Workers did not always have instances of every class. For each image to be collected, they were given ten choices out of which to select one that is available or request ten other choices. This naturally would lead to an extreme class imbalance as the easiest and most common classes would be vastly overrepresented. To make the class distribution more uniform, we presented objects inversely proportional to how frequent they are; the resulting distribution is fairly uniform, see fig. 4.

Objects were described to workers using one to four words, depending on the class. Two exceptions were made, for forks and spoons, as user agreement on how to label two orthogonal faces of these object classes is very low; rough sketches were shown instead. When aligning their object and phone, workers were instructed to ignore the aspect ratio of the rectangular prism. We found that having a single aspect ratio, a cube for example, for all object classes was very confusing to workers. Each object class is annotated with a rough aspect ratio for its rectangular prism. This again represents a small bias toward particular kinds of objects, although this is alleviated by the fact that most objects did not fit a rectangular prism anyway. Deformable objects were still rotated and users followed those rotations aligning the semantically meaningful axes with object parts, but other details of the object pose were not controlled for.

No instructions were given about how to stabilize objects in the desired poses. When necessary, some workers held the objects while other propped them up. For each image, workers were asked two questions on their phone collection UI: to verify that the image depicts an object of the intended class and that it is not too blurry. In many indoor lighting conditions, particularly with low-end cameras, it is easy to take unrecognizable photos without careful stabilization. We estimate the task took around 1.5 minutes per object on average and workers were paid 10 dollars per hour on average.

In total, 95,824 images were collected from 5,982 workers out of which 50,000 images were retained after validation and included in the dataset. Each image was manually verified. About 48% of the data collected was removed. In 10% of images, objects were placed in incorrect backgrounds, showed faces (0.2% of images), or contained other private information (0.03% of images). We found that despite instructions, many users took photos of screens if they did not have an object (23%) – these were removed because on the whole they are very easy for models to recognize. Centralized locations that employ workers on Mechanical Turk were eliminated from the dataset to ensure that objects are not imaged on the same backgrounds across many workers (20%). Note that some problem categories overlapped. So as not to bias the dataset toward images which are easy for humans, validators were instructed to be permissive and only rule out an image of an object if it clearly violated the constraints. Since workers who carry out the task correctly do so nearly perfectly, while workers who do not, carried out almost every trial incorrectly, we have additional confidence that images which are hard to recognize depict the correct object classes.

This dataset construction method is not without its limitations. All objects are indoor objects which are easy to manipulate, they cannot be too large or small, fixed to the wall, or dangerous. We cannot ask workers to manipulate objects in ways that would damage or otherwise permanently alter them. Some object classes which are rare can be difficult to gather and are more likely to have incorrect images before validation. Not all undesirable correlations are removed by this process; for example, some objects are more likely to be held than others while certain object classes are predisposed to have particular colors. We are not guaranteed to cover the space of shapes or textures for each object class. Finally, not all object classes are as easy to rotate, so the resulting poses are still correlated with the object class.

## 4  Results

We investigate object detector performance on ObjectNet using an image labeling task; see section 4.1. Then we explain this performance by breaking down how controls affect results; section 4.2. Finally we demonstrate that the difficulty of ObjectNet lies in the controls, and not in the particular properties of the images, by fine-tuning on the dataset; section 4.3.

### 4.1  Transfer from ImageNet

We tested six object detectors published over the past several years on ObjectNet, choosing top performers for each year: AlexNet (2012) [4], VGG-19 (2014) [5], ResNet-152 (2016) [6], Inception-v4 (2017) [7], NASNET-A (2018) [8], and PNASNet-5L (2018) [9]. All detectors were pre-trained

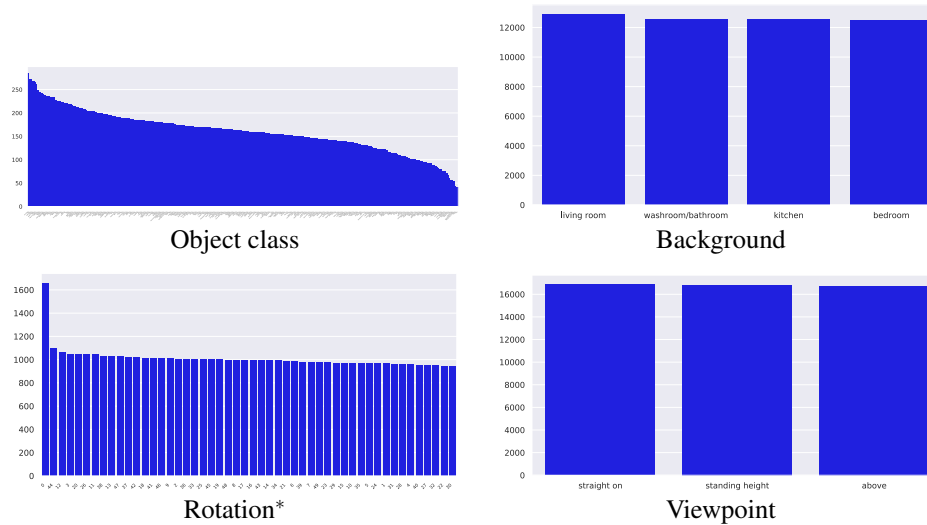

Figure 4: The distribution of the 313 object classes, backgrounds, rotations, and viewpoints in the dataset. The class distribution is fairly uniform due to biasing workers toward low-frequency objects. Object backgrounds, viewpoints, and rotations were sampled uniformly but rejected data can skew the distribution. Each image is also labeled with a 3D rectangular prism and semantically meaningful faces for each object. Spherical objects pop out of the rotation histogram as they have a single rotation. (*) Note that object rotations are less reliable than this indicates: not all objects are equally easy to rotate, the actual rotations of objects pictured in the dataset are less uniform. This represents the object rotations that workers were asked to collect. While this is also true for background and viewpoint, we expect that the true rotation graph is more skewed than the other two.

Air freshener, *Alarm clock*, *Backpack*, Baking sheet, *Banana*, *Bandaid*, Baseball bat, Baseball glove, *Basket*, Bathrobe, *Bath towel*, Battery, Bed sheet, *Beer bottle*, Beer can, Belt, *Bench*, *Bicycle*, Bike pump, Bills (money), *Binder (closed)*, Biscuits, Blanket, Blender, Blouse, Board game, Book (closed), Bookend, Boots, *Bottle cap*, Bottle opener, Bottle stopper, Box, Bracelet, Bread knife, *Bread loaf*, Briefcase, Brooch, *Broom*, *Bucket*, *Butcher's knife*, Butter, Button, CD/DVD case, Calendar, *Can opener*, *Candle*, Canned food, *Cellphone*, Cellphone case, Cellphone charger, Cereal, *Chair*, Cheese, Chess piece, Chocolate, Chopstick, *Clothes hamper*, Clothes hanger, Coaster, Coffee beans, Coffee grinder, Coffee machine, Coffee table, Coin (money), Comb, *Combination lock*, *Computer mouse*, Contact lens case, Cooking oil bottle, Cork, Cutting board, DVD player, Deodorant, *Desk lamp*, Detergent, *Dishrag or hand towel*, Dish soap, Document folder (closed), Dog bed, *Doormat*, Drawer (open), *Dress*, Dress pants, Dress shirt, *Dress shoe (men)*, Dress shoe (women), *Drill*, *Drinking Cup*, Drinking straw, Drying rack for clothes, *Drying rack for plates*, Dust pan, Earbuds, Earring, Egg, Egg carton, *Envelope*, Eraser (white board), Extension cable, Eyeglasses, *Fan*, Figurine or statue, First aid kit, Flashlight, Floss container, Flour container, Fork, *French press*, *Frying pan*, Glue container, Hair brush, Hair clip, *Hair dryer*, Hair tie, *Hammer*, Hand mirror, Handbag, Hat, Headphones (over ear), *Helmet*, Honey container, Ice, Ice cube tray, *Iron*, Ironing board, Jam, Jar, *Jeans*, Kettle, *Keyboard*, Key chain, *Ladle*, *Lampshade*, *Laptop (open)*, Laptop charger, Leaf, Leggings, *Lemon*, *Letter opener*, Lettuce, Light bulb, *Lighter*, *Lipstick*, Loofah, Magazine, Makeup, Makeup brush, Marker, *Match*, *Measuring cup*, *Microwave*, Milk, *Mixing/Salad Bowl*, *Monitor*, Mouse pad, Mouthwash, *Mug*, Multitool, *Nail*, Nail clippers, Nail file, Nail polish, Napkin, *Necklace*, Newspaper, Night light, Nightstand, Notebook, Notepad, Nut for a screw, *Orange*, Oven mitts, *Padlock*, *Paintbrush*, Paint can, Paper, Paper bag, Paper plates, *Paper towel*, Paperclip, Peeler, *Pen*, Pencil, Pepper shaker, Pet food container, Landline phone, Photograph, *Pill bottle*, Pill organizer, *Pillow*, *Pitcher*, Placemat, *Plastic bag*, Plastic cup, Plastic wrap, *Plate*, Playing cards, Pliers, *Plunger*, *Pop can*, *Portable heater*, Poster, Power bar, Power cable, *Printer*, Raincoat, Rake, Razor, Receipt, *Remote control*, Removable blade, Ribbon, Ring, Rock, Rolling pin, *Ruler*, *Running shoe*, *Safety pin*, *Salt shaker*, *Sandal*, Scarf, Scissors, *Screw*, Scrub brush, Shampoo bottle, Shoelace, Shorts, *Shovel*, Skateboard, *Skirt*, *Sleeping bag*, Slipper, Soap bar, *Soap dispenser*, *Sock*, *Soup Bowl*, Sewing kit, *Spatula*, *Speaker*, Sponge, Spoon, Spray bottle, Squeegee, Squeeze bottle, Standing lamp, Stapler, Step stool, *Still Camera*, Sink Stopper, *Strainer*, *Stuffed animal*, Sugar container, *Suit jacket*, Suitcase, *Sunglasses*, *Sweater*, *Swimming trunks*, *T-shirt*, *TV*, Table knife, Tablecloth, Tablet, Tanktop, Tape, Tape measure, Tarp, Teabag, *Teapot*, *Tennis racket*, Thermometer, Thermos, Throw pillow, *Tie*, Tissue, *Toaster*, *Toilet paper roll*, Tomato, Tongs, Toothbrush, Toothpaste, Tote bag, Toy, Trash bag, *Trash bin*, Travel case, *Tray*, Trophy, Tweezers, *Umbrella*, USB cable, USB flash drive, *Vacuum cleaner*, *Vase*, Video camera, Walker, Walking cane, *Wallet*, *Watch*, *Water bottle*, Water filter, Webcam, *Weight (exercise)*, *Weight scale*, *Wheel*, Whisk, *Whistle*, *Wine bottle*, Wine glass, *Winter glove*, *Wok*, Wrench, Ziploc bag

Figure 5: The 313 object classes in ObjectNet. We chose object classes that were fairly common, not too similar to one another, cover a wide range of objects available in homes, and can be safely manipulated by workers. The 113 classes which overlap with ImageNet are marked in italics.

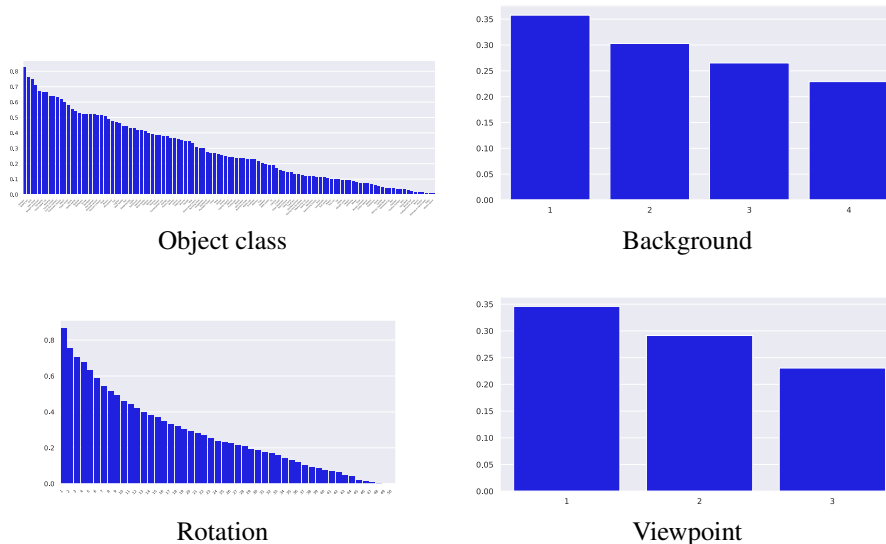

Figure 6: Top-1 performance of ResNet-152 pretrained on ImageNet on the subset of ObjectNet – 113 classes which overlap with ImageNet – as a function of controls used. No fine-tuning was performed; see section 4.3. Classes such as plunger, safety pin and drill have 60-80% accuracy, while French press, pitcher, and plate have accuracies under 5%. Background, rotation, and viewpoint are reranked for each class and then aggregated. All controls have a significant effect on performance and explain the poor performance on the dataset as the disparity between the best and worst performing settings of each of these is 10-20%. The rotation graph is affected by the fact that per-object-class rotations are not uniform. Some per-class rotations are not available, due to the data cleanup phase, meaning that later bins contain few images per class.

on ImageNet and tested on the 113 object classes which overlap between ObjectNet and ImageNet. Performance drops by 40-45% across detectors regardless of top-1 or top-5 metrics; see fig. 1. This performance gap is relative to the performance of detectors on the overlapped classes in ImageNet – our chosen classes were slightly difficult than the average ImageNet class. Increased performance on ImageNet resulted in increased performance on ObjectNet but the gap between the two does not show signs of closing.

## 4.2 The impact of controls on performance

One might wonder about the cause of this lowered performance, even on classes shared with ImageNet. In fig. 6, we break down performance by controls. There is a large gap in performance as a function of background, rotation, and viewpoint. Distributions over these properties were first computed by object class, reranked from highest to lowest performing, and averaged across object classes. If these were irrelevant to detectors and detectors were robust to them, we would see a fairly uniform distribution. Instead there is a large performance gap depending on the background (15%), rotation (20%) and viewpoint (15%). Note that this is despite the fact that we only gave general instructions about backgrounds; we did not ask users where in a room to pose an object and how cluttered the background should be. These together account for much of the performance difference: if one recreates dataset bias by choosing only the better-performing conditions for these controls, object detector performance is mostly restored to that which is seen in ImageNet and other datasets.

## 4.3 Fine-tuning

To emphasize that the difficulty of ObjectNet lies in the controls, and not in the particulars of the data, we – as a one-time exception to the clause which forbids updating parameters on the dataset – fine-tune on the dataset. Kornblith et al. [25] carry out a comprehensive survey on transfer learning from ImageNet to 11 major datasets. On those 11 datasets, training on only 8 images per class increased top-1 accuracy by approximately 37% with variance 11% – only two datasets had less than 30% performance increase because baseline performance was already over 60% with transfer

learning on a single image. We used a ResNet-152, trained on ImageNet, and retrained its last layer in two conditions. The first, using a subset of the ObjectNet classes which overlap with ImageNet. Top-1 performance without fine-tuning is 29%, while with fine-tuning on 8 images, it is 39%, and with 16 images, it is 45%. Far less of an increase than on other datasets despite using only classes which overlap with ImageNet, an easier condition than that investigated by Kornblith et al. [25]. Even using half of the dataset, 64 images per class, one only reaches 50% top-1 accuracy.

This is an optimistic result for detectors as it restricts them to classes which were already seen in ImageNet. The more common fine-tuning scenario is to tune on object classes which do not necessarily overlap the original dataset. Including all 313 ObjectNet classes, yields top-1 accuracies of 23% and 28% for 8 and 16 images respectively. Even using half of the dataset, 64 images per class, top-1 accuracy only reaches 31%, far lower than would be expected given the efficacy of fine-tuning on other datasets. Unlike in other datasets, merely seeing images from this dataset does not allow detectors to easily understand the properties of its objects.

## 5  Discussion

ObjectNet is challenging because of the intersection of real-world images and controls. It pushes object detectors beyond the conditions they can generalize to today. ObjectNet is available at `objectnet.dev` along with additional per-image annotations. Our dataset collection platform is extremely automated, which allows for replacing ObjectNet and recollecting it regularly to prevent overfitting hyperparameters or model structure.

Our preliminary results indicate that human performance on ObjectNet when answering which objects are present in a scene is around 95% across seven annotators. The images which are consistently mislabeled by human annotators are difficult for two primary reasons: unusual instances of the object class or viewpoints which are degenerate. We intend to more carefully investigate what makes objects difficult to recognize for humans as we remove information from the foreground or the background or reduce the viewing time. Predictors for how difficult an image or object is to recognize could see many real-world applications. It is unclear how human-like the error patterns of object detectors are, and if with sufficiently constrained inputs and processing times, human performance might approach that of object detectors.

Aside from serving as a new test set, ObjectNet provides novel insights into the state of the art for object recognition. Detectors seem to fail to capture the same generalizable features that humans use. While steady progress has been made in object recognition, the gap between ObjectNet and ImageNet has remained; since AlexNet no detector has shown a large performance jump. More data improves results but the benefits eventually saturate. The expected performance of many object recognition applications is much lower than traditional datasets indicate. Object detectors are defeated in a non-adversarial setting with simple changes to the object imaging conditions or by choosing instances of objects which appear normal to humans but are relatively unlikely – this makes safety critical applications for object detection suspect. These facts hint toward the notion that larger architectural changes to object detectors that directly address phenomena like those being controlled for here (viewpoint, rotation, and background), would be beneficial and may provide the next large performance increase. ObjectNet can serve as a means to demonstrate this robustness which would not be seen in standard benchmarks.

We find ourselves in a time where datasets are critical and new models find patterns that humans do not, while our tools and techniques for collecting and structuring datasets have not kept up with advances in modeling. Although not all biases can be removed with the techniques presented here, e.g., some materials never occur with certain object classes and some rotations are difficult to achieve, many important classes of biases can. A combination of datasets with and without controls using real-world and simulated data are required to enable the development of models that are robust and human-like, and to predict the performance users can expect from such models on new data.

### Acknowledgments

This work was supported, in part by, the Center for Brains, Minds and Machines, CBMM, NSF STC award CCF-1231216, the MIT-IBM Brain-Inspired Multimedia Comprehension project, the Toyota Research Institute, and the SystemsThatLearn@CSAIL initiative. We would like to thank the members of CBMM, particularly the postdoc group, for many wonderful and productive discussions.

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
