[Reviews · NeurIPS 2019]

Reviewer 1



Overall, the paper is clearly written, with issues and limitations encountered of data collection clearly described. The issue of biased dataset is an important one and it is good to see efforts to tackle this. The main significance of the paper is to point out the importance of controlling for biases in the data used for training machine learning models, and perhaps inspires others to do more controlled data collection. The framework/data is of some interest to researcher working on object recognition. Strengths: - Clearly written and well organized. - The data collection was thorough and careful, with manually verification of final data. Weaknesses: - The controls is limited and missing important variation such as occlusion and clutter. It also introduces biases of its own (for instance, to achieve the different rotations, some of the objects are unnaturally positioned or held by a person). - The experiments conducted on the dataset do not reveal any special insight on how to improve the models or if any of the detectors are more robust than any other. ---- Post rebuttal: I found the rebuttal to be thoughtful. The dataset, and more importantly, careful thinking about biases, is of benefit to the community. I would be happy to see the paper accepted.

Reviewer 2



Possible typo on page 7, section 4.2. “Object detector performance” - did you mean to say “object recognition”? This is an interesting paper. While the observation that ImageNet images are not representative of many task-specific computer vision problems is fairly well-known (Torralba & Efros 2011), this work does an important job of evaluating generalization to a specifically controlled set of image manipulations. By gathering a specifically annotated dataset, this paper also helps reduce overfitting to confounders that we might not want our object recognition models to pay attention to (e.g. background context). One nice experiment in this paper was revealing the breakdown of performance gap induced by background, object rotation, and viewpoint, respectively, and crucially, showing that if the right conditions are chosen, object recognition performance is restored (thus removing the possible explanation that the drop in performance is primarily caused by some other variable such as lighting conditions). ImageNet was constructed in an era where “in-distribution” generalization was quite difficult, i.e. models were not very good even when they had access to the “brittle priors”/contextual confounders. Thus, it does serve its purpose for validating in-distribution “generalization” to a wide variety of classes, even though the evaluation set still comes from “aesthetic images” (Recht et al). Till evidence shows otherwise, I don’t consider ObjectNet to be a *better* evaluation than ImageNet, unless the authors demonstrate that ObjectNet evaluation scores are more accurate by the metrics that ImageNet test accuracy cares about too. One experiment would be measuring ImageNet eval accuracy on a model trained using ObjectNet. Even if this is not a major motivation of the paper, I am still curious what the numbers look like to get a qualitative understanding of the diversity in this dataset. Finally, it remains to be seen whether performing model selection / optimization against this metric results in models that generalize well to object rotations in, say, a factory setting. Software like ARKit (iOS) or ARCore (Android) could be used to assist the crowdsourced human operators in collecting much more accurate object poses / camera poses than relying on the human to position the camera and object accurately.

Reviewer 3



The paper makes an interesting and timely contribution in investigating controlled dataset collection, and the impact of different axes of variation on object detection. In general, the community agrees on the importance of these questions, but there is very little work done to provide answers. As such, the originality and significance of the work is high. Clarity of the paper is also good, and release of the dataset and code should help with reproducibility. *** Post-rebuttal comments After reading the other reviews and the rebuttal, I am more convinced that this paper should be accepted. The rebuttal addressed concerns in a thoughtful and concrete manner.

[Author Response · NeurIPS 2019]

Thank you for the thorough reviews.

1. *New insights about detectors (all reviewers) and which detectors are better (reviewer 1).* At least six new insights:

i) Existing object detectors largely fail to generalize well not because of a generic problem with performance but because of specific phenomena (lack of invariance to background, viewpoint and rotation) that they are brittle to in a way that humans are not. When we selected a subset of images in ObjectNet which had the same biases as those in ImageNet, detector performance on ObjectNet reached that of humans, just as it is in ImageNet. This indicates that focusing on structural methods to handle these phenomena in detectors can have significant payoff.

ii) We demonstrate that there is a large performance gap with respect to humans even over object classes that are part of ImageNet. Human performance over ObjectNet is around 95% on a free-form text answer question, which is considerably higher than networks' performance. This is in stark contrast to other work which indicates that object detectors are approaching human-level performance.

iii) Our results indicate that data alone will not fix these problems. We systematically fine-tuned on ObjectNet using a ResNet-152 trained on ImageNet. Fine-tuning was with 1-64 per class images in steps of powers of 2. Performance increased by $\approx 2\%$ when doubling the data. This shows that even when networks learn biases that lurk in ObjectNet they remain brittle with respect to the controls used.

iv) ObjectNet helps explain why a large gap in performance is seen between developers and users of object detectors in a way that no other modern dataset does. On other datasets fine-tuning recovers the vast majority of the performance very quickly, which is not the case with ObjectNet, and which is not observed, for example, in robotics applications.

v) Large-scale data can be collected in a way that is less biased, not just for object detection but for machine learning in general. Bias is rarely addressed despite affecting all modern machine learning methods.

vi) In a sense, detectors are roughly the same. The gap between ObjectNet and ImageNet is not closing with newer models. 1% gained on ImageNet is also 1% gained on ObjectNet. The community as a whole is making steady measured progress and a breakthrough awaits.

2. *Occlusion and clutter, introduced biases due to people holding objects. (reviewer 1)* Without a doubt, ObjectNet has its own biases which include the fact that certain object shapes are likely to be held when imaged from certain angles. Had we removed more biases, model performance would have likely further decreased. Not all biases are the same though: some are easy to learn and others are difficult. Our fine-tuning results show that learning about how specific objects appear in ObjectNet does not help performance much — far less than the large performance jump seen in other datasets. What we do show is the importance of three particular biases that humans are known to be extremely resilient to from an early age, but networks fail to understand. These overwhelm other types of biases that might help performance. The reviewer astutely points out that we do not consider occlusion or clutter. In the next version of the dataset we intend to do so.

3. *Real-world biases are not necessarily bad, as it can reflect real-world distributions. These biases exist in the world, and leveraging them is useful for task performance in many cases. (all reviewers)* Is it true that real-world biases can be useful and they do increase average-case performance. At the same time, relying on them can be very dangerous. Applications of computer vision to robotics, autonomous cars, surveillance, and other domains require high performance in unusual situations. Indeed, that's where robustness is most needed because those are likely to be situations that were not encountered during the development phase. Additionally, many applications of computer vision can have adversarial opponents — humans that attempt to fool systems by defacing lane markings or hiding dangerous objects from detectors. Most datasets that exist today, ObjectNet being an exception to this, focus on the usual case and do not shed light on the performance one will encounter in unusual cases.

4. *Limitations (all reviewers), simulation and AR (reviewers 2 and 3).* ObjectNet like all methods has many limitations: it consists of indoor objects, the objects have to be available to many people, objects must be mobile, must not be too large or small or fragile or dangerous. We cannot ask users to damage, paint, cut, or otherwise permanently change objects.

In simulation one can create datasets that minimize bias although current methods produce images which are on the whole easy for object detectors even when bias is removed. Foreground/background separation is easier in rendered images and there are fewer sources of noise. In addition, there is a kind of variety and bias that is not easy to control in simulation: shape. There are many possible shapes for a chair but in simulation one will have access to only a small number of variations of static models. Methods exist to synthesize objects of different shapes for the same class but these generally require specifying class boundaries or generative shape models — itself an open and difficult problem.

AR can be useful for collecting data and will play a role as we ask users to pose more complex scenes, like those which include occlusion. We are excited to bring an awareness of more systematic experimental design methods, and show how these can identify the most important phenomena for different machine learning tasks, can help characterize the state of models and compare models in more fine-grained ways, and ultimately help develop new types of models.

[Meta-Review · NeurIPS 2019]

The authors identify and describe problems with biases in data used to train current machine learning systems, introducing a crowdsourcing platform to college a large dataset of object from many different views. They also introduce ObjectNet, 40k crowdsourced images that can be used as a test set for object recognition with variation in object rotation, viewpoints, backgrounds. Evaluation demonstrates that the ImageNet dataset is not sufficient for learning models that are robust to these kinds of object variations. This is a strong paper with several strong contributions